# The Interplay between Autophagy and Virus Pathogenesis—The Significance of Autophagy in Viral Hepatitis and Viral Hemorrhagic Fevers

**DOI:** 10.3390/cells11050871

**Published:** 2022-03-03

**Authors:** Dominika Bębnowska, Paulina Niedźwiedzka-Rystwej

**Affiliations:** Institute of Biology, University of Szczecin, Felczaka 3c, 71-412 Szczecin, Poland

**Keywords:** autophagy, viral hepatitis, viral hemorrhagic fever (VHF), animal model, *Lagovirus europaeus*, rabbit hemorrhagic disease

## Abstract

Autophagy is a process focused on maintaining the homeostasis of organisms; nevertheless, the role of this process has also been widely documented in viral infections. Thus, xenophagy is a selective form of autophagy targeting viruses. However, the relation between autophagy and viruses is ambiguous—this process may be used as a strategy to fight with a virus, but is also in favor of the virus’s replication. In this paper, we have gathered data on autophagy in viral hepatitis and viral hemorrhagic fevers and the relations impacting its viral pathogenesis. Thus, autophagy is a potential therapeutic target, but research is needed to fully understand the mechanisms by which the virus interacts with the autophagic machinery. These studies must be performed in specific research models other than the natural host for many reasons. In this paper, we also indicate *Lagovirus europaeus* virus as a potentially good research model for acute liver failure and viral hemorrhagic disease.

## 1. Introduction

Autophagy is an evolutionarily conserved catabolic process that is necessary for the maintenance of cellular homeostasis in response to the microenvironment, but also physiologically. Autophagy involves selective and non-selective mechanisms leading to the degradation of intracellular substrates, such as damaged cytoplasmic organelles, protein aggregates, or pathogens, in response to changes in the cell’s microenvironment [1]. The initiation and regulation of autophagy are mediated by autophagy-related proteins (ATG). ATG proteins are activated as a result of cellular stress related, for example, to progressive invasion and replication of a virus, resulting in the activation of non-specific and specific immune mechanisms [2]. Autophagy can influence innate immunity mechanisms by modulating signaling associated with pattern recognition receptors (PRRs) and damage-associated molecular patterns (DAMP). Additionally, it has been shown that this process influences the intracellular recognition of pathogens, which facilitates the targeting of bacteria and viruses to autophagosomes [3]. The influence of autophagy on adaptive immunity refers to the delivery of peptides for antigen presentation via major histocompatibility complexes (MHC) located on T cells [4]. Xenophagy is a selective form of autophagy targeting viruses. In this process, intracellular pathogens (bacteria, fungi, parasites, and viruses) are specifically recognized and rapidly directed to the autophagic degradation pathway. It has been shown that some viruses can use the autophagosome to their advantage, since this structure is a protective envelope for daughter virions, and the resulting metabolites and energy are used for the replication process [5]. Thus, many conducted studies focus on the use of autophagic flux as a therapeutic target.

*Lagovirus (L.) europaeus*/GI.1 is a virus that causes severe disease in rabbits (*Oryctolagus cuniculus*) called rabbit hemorrhagic disease (RHD) [6]. *L. europaeus* is a virus belonging to the Lagovirus genus within the *Caliciviridae* family, which consists of non-enveloped ssRNA (+) viruses [7]. It first appeared in China in 1984 [8] and has been global in scope ever since [6]. In 2010, *L. europaeus*/GI.2 [9] appeared, which can induce RHD in rabbits and other species of the Lagomorpha order [10]. Hepatocytes are the target site of *L. europaeus*/GI.1 and GI.2 virus replication, therefore the most spectacular pathologies are observed in the liver [6]. For this reason, studies on *L. europaeus* are successfully used in research on acute liver failure (ALF) [11]. Moreover, several features are observed which are analogous to RHD and viral hemorrhagic fever (VHF) in other species. Therefore, the research on *L. europaeus*/GI.1 and GI.2 may be used as a research model for VHF.

The phenomenon of autophagy is described in viral hepatitis, such as hepatitis B virus (HBV), hepatitis C virus (HCV), but also hepatitis D virus (HDV) [12,13,14]. Nevertheless, it has been reported relatively recently that autophagy is also induced in the course of infections with human hemorrhagic fever viruses from various taxonomic groups. Thus, this process occurs during infection with agents from the family: *Filoviridae*: Ebola virus (EBOV) and Marburg (MARV), *Arenaviridae*: Junin virus (JUNV), *Bunyaviridae*: which includes Old World hantavirus—Hantaan virus (HTNV), and New World hantavirus—Sin Nombre virus (SNV), Crimean–Congo hemorrhagic fever virus (CCHFV), and Rift Valley fever virus (RVFV), but also *Flaviviridae*: dengue virus (DENV) and West Nile virus (WNV) [15,16,17]. Recent data indicate that the process of autophagy is also observed in infection with the *L. europaeus*/GI.1 virus.

In the first part of our work, we summarize the current state of knowledge on the use of *L. europaeus* as an animal model for both ALF and VHF. Next, a review of the available studies on autophagy in viral hepatitis and VHF indicates a need for further research to enable the future therapeutic management of autophagic flux in patients. Research on autophagy induced by a viral infection in the natural host may be a key element in the discovery of so far unknown mechanisms, that may prove helpful in the issues of ALF and VHF. 

## 2. *Lagovirus europaeus* Is a Study Model for ALF

Acute liver failure (ALF) or fulminant hepatic failure (FHF) is a condition characterized by encephalopathy and coagulopathy with organ damage due to apoptosis and necrosis. Renal failure is also a frequent symptom associated with a poor prognosis [18]. The diagnosis of ALF is based on the analysis of biochemical and hematological parameters that indicate a malfunction of hepatocytes. The primary clinical parameter of ALF is the Fischer ratio (molar ratio of branched-chain amino acid concentration [ACR] to aromatic amino acid concentration [AAA]). Specifically, this parameter decreases with the severity of hepatic symptoms [19].

Drug-induced injury is the main cause of ALF in developed countries, while acute forms of viral hepatitis have been reported in developing countries [20,21]. In recent years, a decrease in ALF cases associated with viral infection has been observed due to, among other things, an effective vaccination program and better control of blood products [22]. Most cases of ALF associated with viral hepatitis worldwide are reported after infection with hepatitis A virus (HAV) and hepatitis E virus (HEV) [21]. Hepatitis C virus (HCV) infection is described as the most common cause of non-A and non-B (NANB) hepatitis and can cause fulminant liver failure with a mortality rate of up to 92% [23]. Acute-chronic liver failure (ACLF) is a relatively new clinical entity that refers to the acute worsening of pre-existing chronic liver disease. ACLF is usually accompanied by hepatic encephalopathy, hepatorenal syndrome, and hemorrhagic complications. Guo et al. [24] showed that bleeding was a common complication in patients with HBV-related acute-chronic liver failure (ACLF) and significantly affected the mortality rate, as patients with hemorrhagic complications had a lower 90-day survival rate. Patients with chronic HBV infection may additionally develop HDV superinfection, which may also result in ALF. Furthermore, HDV co-infection can occur in association with acute hepatitis B, resulting in severe or fulminant hepatitis [25]. Additionally, in over 90% of cases, co-infection can evolve into a rapidly progressive chronic infection [26].

RHD has been well characterized as an experimental model in research on ALF [11,27]. In the course of *L. europaeus*/GI.1 and GI.2 infection, several similarities to ALF are noted in terms of clinical symptoms and anatomopathological changes. RHD is manifested by severe hepatitis with necrosis, disseminated intravascular coagulation (DIC), and high mortality, even up to 90% within 48–72 h [27,28]. Tunon et al. [11] described that *L. europaeus*/GI.1 infection meets all the criteria that an animal model of ALF should meet.

The conducted biochemical analyses showed hypoglycemia as well as an increase in blood transaminases, lactate dehydrogenase, and bilirubin in infected rabbits [11,29]. There were also signs of coagulopathy—decreased activity of factor V and factor VII and prolonged prothrombin time. There was also an increase in the concentration of aromatic amino acids in the blood, accompanied by a decrease in the Fischer index. The histopathological picture of the liver showed apoptotic and necrotic areas with neutrophil infiltration and hemorrhages [11,28,29]. Tunon et al. [11] also reported that there is an oxidative–antioxidant imbalance in this infection, and apoptosis is induced by both internal and external signaling pathways. The *L. europaeus*/GI.1 and GI.2 infection provides a good model for ALF because, in addition to the high concordance of biochemical and clinical features of ALF, including the occurrence of intracranial hypertension and encephalopathy, it is also characterized by the reproducibility of test results, as well as the size of the animal allowing for the collection of adequate amounts of material for testing. Moreover, there is a sufficiently wide therapeutic window in the infection, and the pathogen is safe for human health and life [11,27].

## 3. *Lagovirus europaeus*—A Potential Model for Viral Hemorrhagic Fever

Viral Hemorrhagic Fever (VHF) is a term referring to a group of life-threatening infections caused by several virus families (*Arenaviridae*, *Bunyaviridae*, *Filoviridae*, and *Flaviviridae*) [30]. Human infection can occur through contact with vectors or infected people. During the disease, patients may experience a wide range of nonspecific symptoms. Depending on the virus, the infection may have a mild form, while many infections may also be characterized by an acute course with fever accompanied by hypervolemia and coagulopathy, resulting in bleeding and shock [31]. A difficult diagnostic process, lack of effective treatments, and problems with vaccination mean that VHF remains a major challenge in public health, particularly in subtropical regions. Several factors make it inefficient to analyze VHF in humans, including high virulence and rare outbreaks occurring in geographically remote areas with poor diagnostic facilities **[30]**. For this reason, animal models are the best source for advancing our knowledge of VHF because they allow for the advancement of research analyzing disease pathogenesis, therapeutic strategies, and vaccine efficacy. 

It has been described that a good VHF model should meet several criteria: induced primary infection of monocytes/macrophages and dendritic cells; uncontrolled expansion of the virus into multiple organs; viral suppression of type I interferon responses; triggering the secretion of large amounts of pro-inflammatory cytokines; and liver damage due to infection of Kupffer cells and hepatocytes [32]. 

*L. europaeus*/GI.1 and GI.2 infection can occur in three clinical forms (hyperacute, acute, and subacute) and is accompanied by fever > 40 °C. In the course of RHD, nervous system symptoms are observed, including even coma in extreme cases. In addition, gastrointestinal (diarrhea or constipation) and respiratory (difficulty breathing, dyspnea) disturbances are noted. Hemorrhagic nasal and vaginal secretions and markedly bloodshot conjunctivae accompanied by watery eyes are also observed [6,33]. Impaired coagulation by prolonged prothrombin time, increased fibrin degradation, and decreased antithrombin III activity results in the formation of microemboli and the development of disseminated intravascular coagulation (DIC), which is associated with multiple hemorrhages and embolisms in multiple organs [34]. The target cells for viral replication are hepatocytes, which is why acute hepatitis due to virus-induced apoptosis is found in autopsy, as well as splenic enlargement [35]. Macrophages and monocytes of the lungs, lymph nodes, and monocytes found in hepatic vessels are also targets of the virus [36]. Kupffer cells have been reported to be involved in viral replication and are presumably involved in the spread of viral particles in the body [37]. Infection is also characterized by an impaired immune response due to apoptosis of T and B lymphocytes in the liver and peripheral blood, leading to severe leukopenia [38,39]. Cytokines play an important role in modulating the immune response. Changes in the activity of many pro-inflammatory and anti-inflammatory cytokines have also been reported in infection [40,41,42], which perhaps leads to a cytokine storm that is common in some VHF. 

Based on the available literature data, it seems that *L. europaeus*/GI.1 and GI.2 infection shares many common features with VHF, making it a good research model for other hemorrhagic fevers. In addition, and most importantly, it meets most of the described criteria for a good animal model of VHF. 

All similarities between RHD due to *L. europaeus* virus infection, viral hepatitis and VHF infection are shown in Figure 1.

## 4. Characteristics of the Autophagy Pathway

Due to the method of delivering unnecessary or harmful material for lysosomal degradation, there are three paths of autophagy: microautophagy, chaperone-mediated autophagy (CMA), and the best-known form of autophagy—macroautophagy [43,44]. Microautophagy is characterized by the fact that the charge is directly captured by cavities in the lysosomal membrane, while dissolved cytosolic proteins are transported into vesicles of late endosomal multivesicular bodies in endosomal microautophagy [45]. In the case of CMA, the material is directed for degradation via cytoplasmic chaperone proteins which mediate the uptake of unfolded proteins that is dependent on lysosomal membrane proteins 2A (LAMP2A) [5,43]. In turn, macroautophagy (hereinafter called autophagy) is accompanied by sequestration of cytoplasmic proteins and damaged organelles in de novo vesicles—autophagosomes [46]. The resulting autophagosome combines with the lysosome, which results in the formation of an autolysosome—the target formation in which substrates are degraded by lysosomal hydrolases. It is also possible to fuse the autophagosome with early or late endosomes, which results in amphisomes’ formation, which can combine with lysosomes to form autolysosomes [44,47].

Autophagy can be a non-selective, but also a selective process—where the load is recognized by receptors that enable specific identification of the material and lead to its degradation. Moreover, autophagy can be divided into constitutive and induced [43,48]. Autophagy is a cytoprotective process, therefore the maintenance of cell homeostasis requires that the autophagic activity be kept constitutively low in the cell all the time [47,49]. Upregulation of autophagy genes occurs as a result of the action of many different stress-inducing stimuli, which include infectious states [50].

The autophagy process is characterized by separate stages (Figure 2). The main coordinators of most of these stages are ATG proteins. Autophagy begins with the activation of the Unc-51-like autophagy activating kinase (ULK) molecular complex consisting of ULK-1/2, ATG13, ATG101, and the FAK family-interacting protein of 200 kDa (FIP200). The activity of this complex is under the control of the mammalian target of rapamycin kinase (mTOR) [51,52]. Under normal conditions, mTOR negatively regulates the expression of autophagic genes by inactivating ULK. In turn, as a result of stress stimuli, mTOR dissociates from ULK, causing its activation [53].

In the ULK complex, the ATG13 is a signal carrier for autophagy initiation [53]. ATG13 activates and stabilizes ULK1/2 and mediates the interaction of ULK1/2 with FIP200 as it facilitates the phosphorylation of FIP200 by ULK1/2 [54]. The ATG101 protein is located on the phagophore, and its role is to stabilize the expression and the basic level of phosphorylation of the ATG13 and ULK1/2 proteins [51,55,56]. Induction of autophagy is followed by a nucleation process in which the class III phosphatidylinositol 3-kinase (PtdIns3K) complex plays a key role. The PtdIns3K complex consists of phosphatidylinositol 3-kinase VPS34 (VPS34), Beclin-1, p150, and ATG14, and it enables the formation of the PI3P phospholipid [47,48]. Then, the PtdIns3K kinase transfers phospholipid PI3P and phosphatidylinositol 3-phosphate (PtdIns3P) to the site of isolation membrane formation. Then, PtdIns3P is bound by various effectors, including mammalian WIPI (WD repeat domain phosphoinositide-interacting protein 1), which are essential for phagosome formation. These binding proteins form at the edge of the phagophore, another important complex with ATG2 (ATG2-WIPI) which is necessary for the expansion of the phagophore. This complex is recruited at the point of contact between the phagophore being formed and the exit site of the endoplasmic reticulum (ERES). At this point, ATG2 is presumably involved in the transfer of building phospholipids from ERES to the expanding phagophore [57]. Moreover, this stage of autophagy also includes the ATG9 membrane protein located at the phagophore assembly site, which presumably supplies proteins and lipids to the forming autophagosomal membranes [58]. The ATG9 protein also interacts with ATG2, which negatively affects the localization of the ATG2-WIPI complex at the edge of the nascent phagophore [57]. The activity of the PtdIns3K complex can be regulated by several proteins [47,59]. For example, the positive regulators of this complex include UV Radiation Resistance Associated (UVRAG), which increases the activity of PtdIns3K kinase. On the other hand, the Rubicon protein can be included among the negative regulators of autophagy [48,60,61].

The formation of the autophagosome is dependent on two ubiquitin-like protein systems. The first complex forms the conjugated ATG12 and ATG5, which then combine with ATG16L, resulting in the ATG12-ATG5-ATG16L complex formation, making it possible to extend the isolating membrane and thereby form an autophagosome. The formation of the ATG12-ATG5-ATG16L complex is conditioned by the action of activating ATG7 and conjugating ATG10 [43]. After the mature autophagosome is created, the ATG12-ATG5-ATG16L complex dissociates from the membrane [47,62]. The second complex necessary for autophagosome formation is microtubule-associated protein 1A/1B-light chain 3- phosphatidylethanolamine (LC3-I-PE). LC3 is the most described member of the ATG8 protein family. In mammals, the ATG8 orthologs are divided into two subgroups: LC3 and GABARAP. LC3 is formed as a pro-LC3 precursor and is converted to the cytosolic form LC3-I by proteolytic cutting by ATG4, which in mammals occurs in four different isoforms [43]. Through the action of ATG7 and ATG3, the LC3-I form is fused to phosphatidylethanolamine (PE) and transforms into the mature form LC3-II, which is incorporated into the insulating membranes of autophagosomes [45,50,63]. Moreover, the concentration of LC3-II correlates with the number of autophagosomes produced and is, therefore, a common marker of autophagy. Subsequently, LC3-II binds to the autophagic p62/sequestosome 1 (SQSTM1) protein [64]. This factor belongs to a family of proteins called the cargo receptors or selective autophagy receptors (SARs) that recognizes ubiquitin located on the target material to be degraded, creating a platform between the autophagosome and the discharged load. However, p62/SQSTM1 is the best-known member of the cargo receptor family [65]. 

Total autophagic flux requires an autophagosome maturation stage. This process involves fusing with endosomes or lysosomes to form autolysosomes where charge degradation occurs. The autophagosome–lysosome fusion process is coordinated by transmembrane-soluble N-ethylmaleimide-sensitive factor attachment protein receptor (SNARE) proteins. Among the SNARE family, syntaxin 17 (STX17), synaptosomal-associated protein 29 (SNAP29), and vesicle-associated membrane protein 8 (VAMP8) proteins are essential for the formation of the autolysosome [66]. Moreover, the SNARE proteins not only provide autophagosome–lysosome fusion but are also involved in the assembly of precursor vesicles prior to phagophore formation [67].

## 5. Studies on Autophagy in *Lagovirus europaeus*/GI.1 and GI.2 Infection

Studies analyzing the occurrence of autophagy during the experimental infection with *L. europaeus*/GI.1 were performed [68]. Thus, transmission electron microscopy (TEM) analysis showed an increase in the number and content of autophagic vesicles in the livers of infected rabbits, and these structures were observed starting at 12 h post-infection (p.i.). Additionally, the activity of proteins crucial for the autophagy process, such as LC3, was assessed in the liver. It was found that this marker was detected in hepatocytes already after 12 h p.i., and the peak activity was after 24 h p.i. [68]. The conversion of cytosolic LC3-I to lipidized autophagosome-related LC3-II was also recorded. The significant increase in activity which occurred at 18 and 24 h p.i. undeniably confirmed the formation of autophagosomes. Molecular analyses showed early and increased expression of the Atg12-Atg5-Atg16L1 complex. It was recorded that Atg12 and Atg5 levels increased starting at 12 h p.i., and for Atg16L1, they increased from 18 h p.i. and for all markers, peak values occurred at 18 h p.i [68]. Additionally, upregulation of the Beclin-1 gene was detected in 18 and 24 h p.i. The upstream regulator for the Beclin-1- PtdIns3K pathway is UVRAG, for which the maximum value was recorded at 24 h p.i [68]. The experiment also assessed p62/SQSTM1 expression, which is a common marker of autophagic flux. Thus, the analysis showed that the value of p62/SQSTM1 increased starting at 12 h p.i. [68]. The impact of *L. europaeus*/GI.1 infection on the regulatory pathway of autophagy–mTOR, was also assessed. It was shown that phospho-mTOR expression increased between 12 and 24 h p.i., indicating stimulation of the mTOR pathway in parallel with the development of autophagy. These data indicate that mTOR does not down-regulate autophagy in *L. europaeus*/GI.1 infection [68]. Studies [69] have shown that endoplasmic reticulum (ER) stress in *L. europaeus*/GI.1 infection can be induced by modulation of the three branches of the unfolded protein response (UPR). Activation of UPR in rabbits with RHD was evaluated by determining the stress response marker ER—C/EBP homologous protein (CHOP) and glucose-regulated protein (GRP94), which is a protein involved in protein folding, thus the peak expression of these chaperones was determined at 24 h p.i., which occurred in parallel with the increase in Beclin-1, Atg12-Atg5-Atg16L1, and LC3 [68]. This suggests that ER stress can induce autophagy, and additionally it is confirmed by the upregulation of Beclin-1 in *L. europaeus*/GI.1 infection because expression of this protein is required for ER stress-induced autophagy [68]. Studies have been conducted [70] to determine whether the administration of melatonin modulates the autophagic response in *L. europaeus*/GI.1 infected rabbits. The analysis of the results showed a decreased expression of the LC3-II protein as well as Beclin-1, Atg5, Atg12, and Atg16L1, which confirmed that melatonin causes partial inhibition of autophagy in RHD.

So far, no studies have been conducted on autophagy in *L. europaeus*/GI.2 infection, therefore a more in-depth understanding of the autophagy status in the course of RHD requires further analysis.

## 6. Autophagy in Viral Hepatitis

### 6.1. Hepatitis B Virus

HBV is a virus belonging to the *Hepadnaviridae* family, and its genome is a dsDNA-RT molecule that encodes four genes: S, X, C, and P. Gen S encodes HBs surface proteins —short (SHB), medium (MHB), and large (LHB), gene X produces a product in the form of nonstructural HBx protein that affects viral replication, gene C encodes HBcAg core antigen, while viral polymerase is encoded in the P gene [71]. According to current knowledge, autophagy is involved in the HBV life cycle. In particular, stimulation of autophagy promotes viral replication; however, the mechanisms by which HBV modulates the autophagic pathway are not clear.

Studies indicate that HBx protein can induce autophagy and may promote the pro-autophagic Beclin-1 expression [62]. Additionally, the HBx protein can bind to VPS34 and this condition leads to autophagy to promote viral replication [72]. Similarly, Son et al. [73] found that HBx enhances the interaction between VPS34 and the TNF Receptor Associated Factor 6 (TRAF6)-Beclin-1 complex, thereby increasing Beclin-1 ubiquitination and disrupting Beclin-1-BCL-2 complex formation, leading to increased induction of autophagy. It has also been shown [74] that HBx protein dephosphorylates death-associated protein kinase (DAPK) leading to its activation in the Beclin-1-related pathway, but contrary to other studies [75] it did not induce autophagy through the c-Jun N-terminal kinases (JNK) pathway.

Indeed, many studies indicate that HBx is involved in autophagy induction and autophagy machinery [2,71,72,73,76]. However, data indicate that HBx leads to incomplete autophagy to promote viral replication [74,75,76,77,78]. In the study of Liu et al. [76], it was due to acidification of lysosomes without disrupting autophagosome–lysosome fusion, which impaired cargo degradation, and resulted in more autophagosomes. HBx also interacts with the c-Myc, which inhibits the expression of microRNA–miR-192-3p, which targets X chromosome-associated apoptosis protein (XIAP) inhibitors [77]. It has been described that HBV promotes autophagy early in the process through the miR-192-3p-XIAP pathway involving nuclear factor kappa-light-chain-enhancer of activated B cells (NF-κB), leading to the upregulation of Beclin-1 in vitro and in vivo. 

On the other hand, in the study of Li et al. [78], autophagic flux in hepatoma cells was not dependent on HBx but viral SHB protein and the reasons for this discrepancy are not entirely clear. SHB protein triggers ER stress, which induces autophagy via pathways other than HBx and affects viral envelope amplification affecting only a small part of HBV replication. To alleviate stress, the UPR pathway is activated in the cell, which is coordinated by three stress sensors: inositol-requiring enzyme 1 (IRE1), ER-like RNA protein kinase (PERK), and activation of transcription factor 6 (ATF6). As a result of SHB-induced ER stress, IRE1, PERK, and ATF6 signaling is activated which induces UPR-related autophagy [78]. 

In summary, the effects of HBV on the autophagy machinery are fairly well understood. Analysis of the available results from in vitro and in vivo studies indicates that there are many contradictions within the available data. This implies the presence of a need for further studies in different model systems to fully understand the mechanisms of HBV and autophagy relationships, which may prove crucial for the development of therapeutic strategies.

### 6.2. Hepatitis D Virus

HDV is a virus belonging to the *Hepadnaviridae* family and it is the only member of the Deltavirus genus, whose genome is a (−)ssRNA molecule that encodes a single nucleocapsid protein present in two isoforms: (small) S-HDAg and (large) L-HDAg. Importantly, it is a defective pathogen because it requires the presence of HBV for virion assembly and export, while RNA replication remains autonomous [79].

Research into the relationship between autophagy and HDV is limited. Recent studies on hepatocyte cell lines showed that especially ATG5, but also ATG7 and LC3 knockout, decreased intracellular HDV RNA level, but did not affect HDV release [14]. The same study showed that HDAg promotes autophagosome accumulation to then block autophagic degradation to promote viral replication [14]. 

Given that anti-HBV therapies are ineffective against HDV infection, there is a strong need to explore new therapeutic strategies. Thus, targeting autophagy may prove to be an effective therapeutic target for both viruses, so there is a strong need to continue research.

### 6.3. Hepatitis C Virus 

HCV is a virus belonging to the *Flaviviridae* family and its genome is (+)ssRNA molecule, which encodes a polyprotein precursor, which is then cleaved by proteases into several structural, core, and envelope proteins; E1 and E2 glycoproteins; as well as non-structural (NS) proteins p7, NS2, NS3, NS4Am NS4B, NS5A, and NS5B [80]. The available data show that HCV activates autophagy to promote replication. The mechanism by which HCV leads to the activation of the autophagic cascade may be direct or indirect. Research by Sir et al. [81] showed that HCV induced ER stress, which resulted in the activation of UPR pathway-dependent autophagy, but protein degradation was limited due to the limitation of autophagosome and lysosome fusion.

HCV can also trigger oxidative stress-induced autophagy. The nuclear erythroid 2-related factor 2/ antioxidant response element (Nfr2/ARE) signaling pathway is triggered in the cell to overcome oxidative stress. In the studies of Medvedev et al. [82], HCV disrupts this signaling pathway to promote the persistence of elevated levels of reactive oxygen species (ROS), which, by activating autophagy, supports the release of viral particles. Direct activation of autophagy may take place through the activity of viral proteins that can affect or activate autophagic proteins. Research by Aweya et al. [83] showed that Beclin-1 binds to the viral p7 protein, but this interaction is insufficient to influence autophagy. Another study [84] showed that the NS4B protein is sufficient for the initiation of autophagy. The authors concluded that NS4B could modify the membrane environment via Rab5 and Vps34, thus activating autophagy. On the other hand, Desai et al. [85] showed that autophagy can negatively regulate HCV replication in the presence of interferon-β (IFN-β), leading to autophagic degradation of the HCV core proteins and NS3/4A in the mouse model. It has also been reported [86] that the NS3 protein can bind to the M protein of the immune-related GTPase family (IRGM), which has an affinity for ATG5, ATG10, LC3, and Bax-interacting factor 1 (Bif-1) to induce autophagy, promoting viral infectious particle production. The *NS5ATP9* gene is involved in regulating many cellular processes, including DNA repair, and is targeted and transactivated by the NS5A protein. Quan et al. [87] showed that *NS5ATP9* is required to enhance *BECN1* expression by NS5A protein. Moreover, NS5A protein mediated the regulation of starvation-induced autophagy and played a key role in the progression of embryonic hepatoma [88]. It has been shown that HCV infection can also trigger a selective form of autophagy targeting mitochondria (mitophagy). Studies [89] showed that the viral NS5A protein induces mitochondrial fragmentation, loss of mitochondrial membrane potential, and Parkin translocation to the mitochondria, which leads to complete autophagic flux. This condition may decrease immune signaling and contribute to the pathogenesis of HCV infection.

A summary of the interaction between viral hepatitis and autophagy is presented in Figure 3.

The figure shows the elements by which viruses affect pathways associated with autophagy in host cells. The blue line indicates a direct activating effect of the viral elements. The black line indicates the activation mechanism of the elements involved in the autophagy pathway. The red line indicates the blocking mechanism. The dashed line indicates the dissociation of the factor. HBx directly binds to VPS34, which activates the PtdIns3K kinase complex and leads to the activation of autophagy. HBx also enhances the molecular binding between VPS34 and Beclin-1 and the ubiquitination of Beclin-1 by TRAF6 in autophagy induced by TLR4 stimulation. HBx interacts with Beclin-1 by inhibiting its association with BCL-2, which is a negative regulator of the induction of Beclin-1 dependent autophagy. HBV mediated by HBx increases Beclin-1 expression by regulating its transcription process. HBx activates DAPK, which phosphorylates Beclin-1 and this condition leads to autophagy. HBx interacts with c-Myc to directly inhibit miR-192-3p expression, which negatively regulates XIAP. In turn, XIAP activates the NF-κβ pathway that up-regulates the expression of autophagic genes. Viral SHB triggers UPR-activated autophagy. HDV protein—HDAg increases the lipidation of LC3-I to LC3-II, which promotes the accumulation of autophagosomes. HCV induces stress ER, which activates UPR-dependent autophagy but proteins’ degradation via autophagic flux is limited. HCV infection is associated with increased levels of ROS in the cell. To overcome oxidative stress, Nrf2 is released from its complex with Keap1. Then Nrf2 translocates to the nucleus, where it binds to sMaf and antioxidant response elements (AREs) inducing the expression of cytoprotective enzymes. Elevated levels of ROS in the cell induce phosphorylation of the adaptor p62 on Ser349 (pS(349)62). As a result, pS(349)p62 (p-p62) binds with increased affinity to Keap1 releasing Nfr2 from this complex. In HCV infection, translocation of sMafs from the nucleus to the cytoplasm occurs, where viral NS3 binds sMafs. Under these conditions, Nfr2 associates with delocalized sMafs, which block its entry into the nucleus and inhibit the expression of cytoprotective genes. The p7 protein interacts with Beclin-1 but this reaction is insufficient to activate autophagy. HCV also interacts with IRGM via NS3 protein to induce autophagy. IRGM is an important regulator of autophagy located in the mitochondria. IRGM influences autophagic flux in the initiation/elongation stage by acting on ATG5, ATG10, LC3, and Bif-1. HCV infection also is related to mitophagy. The NS5A protein induces a loss of mitochondrial membrane potential, mitochondrial fragmentation, and Parkin translocation into mitochondria, leading to the induction of mitophagy.

## 7. Autophagy in Viral Hemorrhagic Fever

### 7.1. Filoviridae

Members of the *Filoviridae* family are (−)ssRNA viruses whose genome encodes a nucleoprotein (NP), polymerase cofactor protein (VP35), matrix protein (VP40), glycoprotein (GP), transcription activator (VP30), and second matrix protein (VP24), dependent from RNA, RNA polymerase (L), and nonstructural protein (sGP) [90,91,92,93].

#### 7.1.1. Ebola Virus (EBOV)

EBOV is a virus that causes fatal hemorrhagic fever in humans and other primates. The infection begins with the entry of the pathogen into the host cell through a non-selective uptake process—macropinocytosis [94]. Research indicates that the autophagic pathway is essential to causing infection in the host. Autophagy proteins are involved in the macropinocytosis process and the passing of macropinosomes into the cell. The research of Shtanko et al. [95] on cell lines showed that forms LC3B-I and LC3B-II interact with macropinosomes, with LC3B-II being of key importance for the internalization of macropinosomes and thus for the penetration of the pathogen into the cell. Similarly, removal of *BECN1* and *ATG7* abolished EBOV capture and vesicle formation, which blocked virus entry into the cell. This confirmed that autophagy-related proteins are essential in triggering EBOV infection in the host cells.

Chiramel et al. [91] conducted a study where it was found that Family With Sequence Similarity 134 Member B (FAM134B)-dependent ER-phagy (selective ER autophagy) has a limiting effect on the replication of EBOV in mouse cells, and therefore it may be a potential therapeutic target.

In the picture of Ebola infection, strong leukopenia is observed, and autophagy may be one of its factors. Research by Younan et al. [96] showed that EBOV, despite the presence of viral mRNA and antigens in CD4+ T cells, did not cause productive infection. On the other hand, exposure of T cells to EBOV induced autophagy by ER stress.

Infection with EBOV can induce CMA autophagy in host cells mediated by Bcl-2–associated athanogene 3 (BAG3) protein. This protein is induced under conditions of cellular stress and regulates many cellular pathways, including autophagy and apoptosis. The aforementioned study showed [97] that BAG3-mediated autophagy CMA is induced by interaction with the viral VP40 protein, which contains a specific PPxY sequence, and this reaction inhibits the spread of the virus and thus constitutes the host’s defense mechanism against infection.

#### 7.1.2. Marburg Virus (MARV)

MARV causes severe hemorrhagic fever in humans and susceptible animals [92,93]. Research focusing on the relationship between autophagy and MARV is very limited. Studies have shown that MARV infection induces autophagy in the host as a defense mechanism against the virus. This process is initiated by the BAG3 protein, as a result of contact of the host cells with the viral VP40 protein, which is analogically similar to that in EBOV [97].

### 7.2. Arenaviridae

Members of the *Arenaviridae* family are viruses whose genetic material is an (-)ssRNA molecule, which is divided into two segments: small (S) and large (L). The S segment encodes a glycoprotein precursor (GPC) and a peptide (SSP). There is also a second part, a nucleoprotein (NP), which in turn is the main component of the ribonucleoprotein core (RNP) guiding replication and transcription. The L segment encodes the viral polymerase and the RING protein [98].

#### Junin Virus (JUNV)

JUNV is a pathogen belonging to the New World arenaviruses, which is the etiological factor of Argentine hemorrhagic fever (AHF) in humans—a severe and highly fatal disease. It has been shown that during the early stage of JUNV infection, autophagy is induced in the host organism, which is an essential element for the effective spread of the virus [99,100]. Studies have been conducted to analyze the role of autophagy in JUNV infection using human cell lines [100]. The obtained results showed that autophagosome formation occurs during infection, as indicated by the increase in the level of LC3-II protein. Co-localization of p62/SQSTM1, ATG16, Rab5, Rab7a, and lysosomal-associated membrane protein 1 (LAMP1) with the LC3 protein were also observed, which confirmed that phagosome maturation occurs during infection including fusion with maturity or late endosomes. Additionally, it was found that p62/SQSTM1 is degraded during JUNV infection, which indicates complete autophagic flux [100]. Studies by Roldan et al. [99] using Atg5 or Beclin-1 knockout cells showed that autophagy is a key process for viral replication. The removal of these elements critical for autophagy resulted in a significant decrease in virus replication.

### 7.3. Bunyaviridae

Members of the *Bunyaviridae* family are viruses with a genome in the form of (-) ssRNA, which consists of three segments: small (S) coding for a nucleoprotein (N), medium (M) coding for Gn and Gc glycoproteins, and a large (L) segment, which encodes RNA-dependent RNA polymerase [101].

#### 7.3.1. Hantaan Virus (HTNV)

HTNV is an Old World hantavirus that causes severe hemorrhagic fever with renal syndrome (HFRS) in humans and is vectored in rodents [102]. Research into the autophagy–HTNV relationship is limited. Wang et al. [103] showed that HTNV manipulates autophagic flux for its benefit. In the early stage of infection, the viral Gn protein interacts with Tu Translation Elongation Factor, Mitochondrial (TUFM), and LC3B to induce mitophagy. This condition interferes with MAVS-mediated RIG-I-like receptor (RLR) signaling in the cell and then inhibits the production of interferon, which is known to be involved in the host’s antiviral response. This state provides the virus with favorable conditions for replication and display. Subsequently, viral NP protein can bind to LC3B and Synaptosome Associated Protein 29 (SNAP29), which inhibits Gn-induced autophagy, and can then exert a positive influence on the production of progeny virus. Moreover, in the same study, the Beclin-1 knockdown inhibited HTNV replication, indicating that this pro-autophagic factor is important for viral infection [103]. However, the details of Beclin-1’s role in HTNV infection are unclear, so further research is required.

#### 7.3.2. Sin Nombre Virus (SNV)

SNV is a virus that belongs to the New World hantavirus group. This agent is responsible for the induction of hantavirus pulmonary syndrome (HPS) [104]. Studies on the connection between autophagy and SNV have shown that this virus uses autophagy in its life cycle. Hussein et al. [17] showed that SNV infection and Gn expression induce autophagy in host cells, as evidenced by the autophagy markers LC3-I and LC3-II. Furthermore, autophagy is critical for viral replication. Experimentally down-regulating the expression of *BECN1* and *ATG7* with small interfering RNA (siRNA) inhibited viral replication [17]. In the same study, the role of the viral Gn protein was also examined. Gn is selectively degraded by the host autophagy machinery, which is required for efficient viral replication. Since Gn is required for later stages of the viral cycle (formation of new virions), studies have been conducted to suggest how Gn avoids autophagic degradation at the appropriate time [105]. It has been suggested that this may be due to the accumulation of viral genomic RNA, nucleocapsid protein, and Gc glycoprotein. Next, it leads to the formation of nucleocapsids that induce viral packaging and assembly, preventing Gn degradation. Another possibility is that Gn, once integrated into accumulated nucleocapsids, is hidden from host autophagy adapters. Gn has a binding site for autophagy adaptor proteins, such as the p62/SQSTM1 protein. It is also speculated that there may be a molecule that is a molecular switch during the degradation of Gn by autophagy early in infection and the survival of Gn during the transition to folding and packaging [105].

#### 7.3.3. Crimean Congo Hemorrhagic Fever Virus (CCHFV)

CCHFV is a virus that causes Crimean–Congo hemorrhagic fever in many mammals as well as in some bird species. However, in humans, the infection causes a serious and life-threatening disease with a mortality of 5–30% [106]. Research into autophagy and CCHFV infection is limited. In the studies of Moroso et al. [107], CCHFV caused a slight increase in the level of *ATG5*, *ATG7*, *ATG3*, *ATG12*, and *BECN1* and a significant increase in the level of *MAP1LC3* and *p62/SQSTM1* transcription. Additionally, CCHFV caused massive lipidation of LC3 in hepatocytes independent of ATG5, ATG7, and Beclin-1 and the alternative ATG12-ATG3 pathway. It was also shown that the autophagic pathway was not involved in either promoting or limiting viral replication. The authors indicated that CCHFV causes LC3 lipidation in previously unknown pathways [107]. 

#### 7.3.4. Rift Valley Fever Virus (RVFV)

RVFV is a virus that causes a serious and dangerous disease of many vertebrates, especially in farm animals, but also in humans, where the infection may lead to encephalitis, retinitis, or hepatitis accompanied by hemorrhagic fever [108].

Research into the relationship between autophagy and RVFV is deficient. Studies by Moy et al. [109] showed that Toll receptors or signaling molecules, e.g., MyD88, are necessary for the activation of the autophagy pathway by RVFV in fly and mammalian cells as an antiviral mechanism. In the fly model, autophagy via Toll-7 reduced viral replication and increased survival. Further, in studies on mouse and human cell lines, silencing of autophagic genes (*Atg5*, *Atg7*, and *Becn1*) also resulted in an increase in viral replication, which also confirms the antiviral properties of autophagy in RVFV infection. Moreover, pharmacological activation of antiviral autophagy in flies and mammals had a strong influence on the inhibition of RVFV infection [109]. On the other hand, other authors obtained results that showed that during infection with this virus, a reduced total level of the autophagic protein LC3B is observed [110], which may suggest that the virus may also target the autophagic pathway. There are indeed large indications that the use of therapies that increase autophagy can be an effective therapeutic approach. 

### 7.4. Flaviviridae

Members of the *Flaviviridae* family are viruses with (+)ssRNA genetic material that encodes single polyproteins that cleave into three structural proteins: capsid (C), vestibule and envelope (E), and seven non-structural proteins: NS1, NS2A, NS2B, NS3, NS4A, NS4B, and NS5 [111].

#### 7.4.1. Dengue Virus (DENV)

DENV has four serotypes (DENV1-DENV4) and is transmitted by *Aedes* mosquitoes. DENV infection causes dengue fever in humans, ranging from mild fever to severe hemorrhagic fever and shock syndrome. Many studies have shown that autophagy plays a significant role in DENV infection. Lee et al. [112] were the first to demonstrate that DENV infection activates the mechanism of autophagy in various cell lines. Moreover, this process was conducive to viral replication. DENV can use some elements of autophagy, such as autophagosomes, amphisomes, and autolysosomes, which serve as scaffolds for viral replication and escape the immune system [113]. Amphisomes play the most important role in DENV entry and localization of the components of viral replication. In the case of DENV-2, it was observed that the virus requires amphisomes, while DENV-3 interacts with both amphisomes and autophagolysosomes as sites for its replication complexes [114].

Research by Metz et al. [115] showed that DENV infection increases the autophagic flux to promote viral replication already in the initial stage of infection and induces p62/SQSTM1 degradation (via a proteasomal mechanism) for its benefit. This means that during DENV infection, the virus initially stimulates and then inhibits the overall and selective autophagy by blocking the formation and degradation of autophagosomes by lysosomes and reducing the level of p62/SQSTM1 [115]. Autophagy activated by DENV infection, in addition to its pro-viral role in the viral life cycle, also influences changes in cellular metabolism [116]. Studies have shown [117] that the DENV virus triggers the process of autophagy at all stages of autolysosomal maturation, in various cell types, to catabolize lipid droplets (LD), release free fatty acids (FFA), and generate ATP for DENV replication. Lee et al. [117] were the first to prove that ER stress induced by DENV-2 through UPR signaling pathways increases the activity of autophagy, DENV replication, and pathogenesis both in vitro and in vivo. The authors suggest that targeting ER and autophagy stress molecules could become a potential therapeutic option for patients infected with DENV [117].

Autophagy can regulate lipid metabolism (lipophagy) by modulating the degradation of triglycerides that have accumulated in the cytosolic lipid droplets. Lipophagy is a selective type of autophagy in which intracellular lipid collapses in the form of lipid droplets (LD) are degraded [3]. Research by Samsa et al. [118] showed that in DENV infection, an increased number of LDs containing viral capsid proteins is observed. This suggests that DENV uses lipophagy for nucleocapsid formation as well as for efficient replication. Additionally, DENV induces the activity of 5’AMP-activated protein kinase (AMPK) and this kinase suppresses the mTORC1 factor. Due to this condition lipophagy is not inhibited and viral replication progresses [119]. Moreover, in the studies of Zhang et al. [120] DENV induced autophagy via Lipid droplet-regulating VLDL assembly factor (AUP1) which is localized in the LDs. It happens through the interaction of AUP1 with viral NS4A and NS4B proteins, which triggers AUP1 acyltransferase activity by generating phospholipids, which are essential elements of the membranes involved in lipophagy [120].

Autophagy is involved in the pathogenesis of DENV infection. It has been shown that autophagy in infection, accompanied by the phenomenon of antibody-dependent enhancement (ADE), promotes viral replication [121]. Sub-neutralizing antibodies created as a result of previous DENV infection may facilitate the binding of the virus and the entry into cells with the Fcγ receptor in secondary infection [121]. Huang et al. [122] reported that in DENV infection with ADE, higher expression of *Atg5* and *Atg12* was induced, but an increased amount of autophagosomes was also observed to promote viral replication. Moreover, in the studies by Mateo et al. [123], pharmacological modulation of autophagy was applied in DENV-infected cells. It has been shown that the presence of autophagic machinery, in addition to facilitating the replication of viral RNA, additionally facilitates the maturation of infectious particles. This makes autophagy inhibitors a promising strategy in DENV infection treatment. In conclusion, despite the demonstration that the autophagy process is activated by DENV infection and is a tool of the virus to aggravate the infection in many experimental systems, further research is still needed as the available results are not fully conclusive. Panyasrivanit et al. [124] showed that the biochemical induction of autophagy in monocytes resulted in a significant reduction in DENV replication. Recently, Chen et al. [125] reported that activation of the autophagy pathway through rapamycin treatment resulted in a reduction in viral particles, which proves the antiviral role of autophagy in *Aedes* mosquito cells. Due to these facts, a full understanding of the pathogenesis of the disease and the physiological importance of DENV-induced autophagy should remain the subject of further research.

#### 7.4.2. West Nile Virus (WNV)

WNV is a virus that causes diseases in humans, horses, birds, and other species, the symptoms of which may range from mild fever to a lethal neuroinvasive form [126]. The role of autophagy in WNV infection is controversial. In the studies by Vandergaast and Frederickensen [127], it was found that neither the pathogenic nor the non-pathogenic WNV strain activated autophagy. Moreover, WNV infection did not block autophagosome formation during the autophagy induced by rapamycin, nor virus-independent accumulation of LC3-II in the cell line model. Additionally, inhibition of autophagy via Atg5 or Atg7 depletion did not affect the viral life cycle. On the other hand, Beatman et al. [128] have shown that WNV infection increases LC3B lipidation in a cell line model and primary neuron cultures. Thus, WNV infection caused lysosomal colocalization with autophagosomes, LC3B-II turnover, and autolysosomal acidification. WNV did not cause significant changes in p62/SQSTM1 levels, despite strong activation of autophagy. Therefore, it is indicated that p62/SQSTM1 is reversed or degraded by autophagic flux. Moreover, virus replication was independent of autophagy. Other studies have reported [129] that the use of Tat-Beclin-1 (autophagy-inducing peptide) exerts antiviral activity by limiting WNV replication and reducing mortality in infected mice. Similarly, Kobayashi et al. [130] showed that WNV infection activates autophagy in infected cells, and this process limits virus replication. Depriving the cells of the autophagic factor—Atg5, resulted in a significant enhancement of WNV replication compared to wild-type cells. In addition, studies showed that autophagy inhibited viral multiplication at the stage of genome replication and gene expression. Research by Blázquez et al. [131] brought new insights into the controversy surrounding findings regarding the link between autophagy and WNV infection. Certain naturally occurring amino acid substitutions in viral NS4A or NS4B proteins have been shown to influence the autophagic response by modifying and aggregating LC3. Moreover, induced autophagy was independent of the UPR pathway [131]. Recently, it has also been shown [132] that viral C protein interacts with AMPK and mediates the ubiquitination of AMPK. AMPK is activated during metabolic stress and is a factor that activates autophagy in the cell. Ubiquitinated AMPK is degraded, which leads to the inhibition of autophagy and the accumulation of protein aggregates in cells, which results in autophagy inhibition and the neuropathogenesis of WNV infection [132].

A summary of the interaction between viral hemorrhagic fever and autophagy is presented in Figure 4.

The figure shows the relationship between viral hemorrhagic fevers and elements of autophagy pathways in the host cells. The blue line indicates the positive effect of the viral elements. The black line indicates the activation mechanisms. The red line indicates the blocking mechanism. The dashed line indicates the translocation of the factor. The autophagic proteins Beclin-1, Atg7, and LC3B are involved in the entry of EBOV into the host cell. Ankyrin repeat and FYVE domain containing 1 (ANKFY 1) is a macropinosome-associated protein. Both the LC3-I and LC3-II forms interact with ANKFY 1 on the cell membrane to mediate the internalization of EBOV into the cell. Infection with EBOV induces ER stress in T cells. In response to this stimulus, the UPR response is activated, which leads to autophagy. In EBOV infection, autophagy is activated by stimulation of TLR4 with the viral Gn protein. BAG3 is a protein that is involved in chaperone-mediated autophagy (CMA). BAG3 interacts with the VP40 protein of EBOV and MARV to activate anti-virus CMA. Raf/MEK/ERK is a cascade of mitogen-activated protein kinase (MAPK) and consists of the serine/threonine Raf kinase, MEK kinase, and extracellular signal-regulated kinase 1 and 2 (ERK 1/2). JUNV activates the Raf/MEK/ ERK pathway to promote its replication. The MEK/ERK pathway control autophagy by regulating Beclin-1 transcription. The HTNV Gn protein translocates into the mitochondria where it interacts with the Tu Translation Elongation Factor, Mitochondrial (TUFM), and recruits LC3B thereby promoting mitophagy. By engaging the mitophagy pathway, the immune response mediated by type I IFN (IFN) is stopped due to the degradation of MAVS. At the appropriate moment, viral NPs inhibit Gn-induced autophagosome formation through competitive binding to LC3B. NP also inhibits the fusion of the autophagosome with the lysosome. NP interacts with synaptosomal-associated protein 29 (SNAP29) to block its association with syntaxin 17 (STX17), which is essential for the autophagosome–lysosome junction. The SNV Gn protein is selectively targeted for autophagic degradation to keep a low level of intracellular Gn, which is necessary for efficient SNV replication. CCHFV causes a subtle enhancement of transcription of autophagic genes and lipidation of LC3 in a non-classical pathway. RVFV lowers the level of LC3 in the infected cell. DENV infection causes ER stress. In response to this stress, various UPR pathways are activated that lead to autophagy. One of the UPR pathways results in the dissociation of BCL-2 from Beclin-1, which in turn leads to the activation of autophagic flux. Infection with DENV causes the activation of AMPK. AMPK activates autophagy by inactivating mTOR, which inhibits the induction of autophagy by affecting the ULK 1/2 complex. Additionally, AMPK initiates autophagy by directly activating the ULK 1/2 complex. AUP1 is a membrane protein associated with lipid droplets (LD). The NS4A and NS4B proteins of DENV work together to induce lipophagy. NS4A interacts with AUP1, leading to the activation of AUP1 followed by lipophagy. The WNV NS4A and NS4B proteins confer the ability to modify and cause LC3 aggregation. WNV protein C causes ubiquitination and degradation of AMPK, which inhibits autophagy.

## 8. Conclusions

Almost every stage of viral infection can be controlled or influenced by autophagy, from promoting antigen presentation to regulating inflammatory responses, to the final destruction of the virus. On the other hand, viruses can manipulate the process of autophagy and use it as a strategy to avoid immune responses, leading to replication and release from infected cells, spreading infection in the host organism (Table 1). Each virus may have a unique strategy to exploit or succumb to autophagy, but some similarities are noticeable through taxonomic affiliation or a similar etiology of the disease. Admittedly, the knowledge of the dependence of some viruses and autophagy is fairly well established. In other cases, it is still unclear whether autophagy is an antiviral mechanism or, on the other hand, autophagic elements constitute the machinery for viral replication. However, the details of the mechanisms behind these interactions are still being discovered and more research is needed to fully understand the role of autophagy in viral hepatitis and VHF. Given that there is an ongoing need for more effective therapeutic strategies for viral hepatitis and hemorrhagic fever, the study of autophagy appears to be a promising approach. The available research analyzing the problems of the infections discussed in this paper is based mainly on in vitro systems, and in vivo research is very limited, therefore the search for new research models is very important.

*L. europaeus*/GI.1 has been used successfully as a model for ALF. However, the characteristics of this pathogen indicate that it also meets the postulates of the VHF model. Thus, further autophagy studies using the *L. europaeus*/GI.1 and GI.2 infection model in its natural host may be crucial for discovering new facts and resolving doubts in the relationship between autophagy and viral hepatitis or VHF.

## Figures and Tables

**Figure 1 cells-11-00871-f001:**
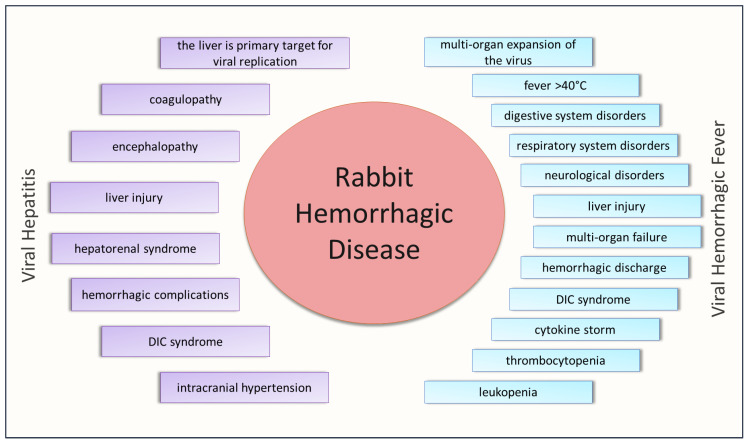
Features of rabbit hemorrhagic disease common to viral hepatitis and viral hemorrhagic fever.

**Figure 2 cells-11-00871-f002:**
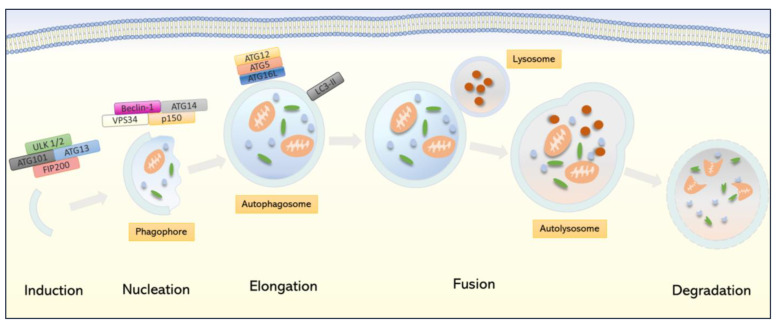
Characteristic of autophagy. The figure shows the pivotal moments in autophagic flux. The key protein components involved in individual phases of autophagy are marked.

**Figure 3 cells-11-00871-f003:**
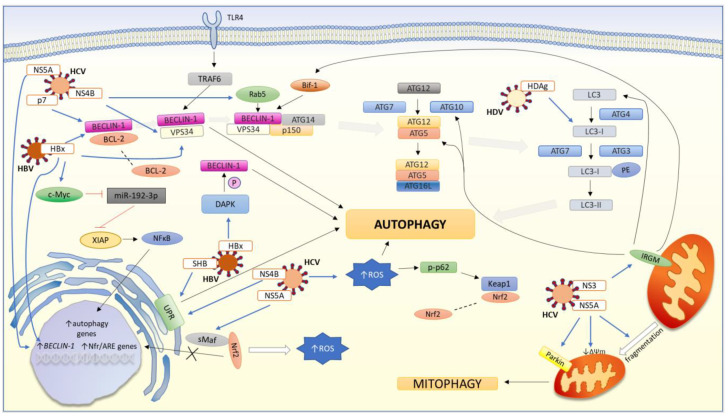
The interaction between viral hepatitis and autophagic pathway.

**Figure 4 cells-11-00871-f004:**
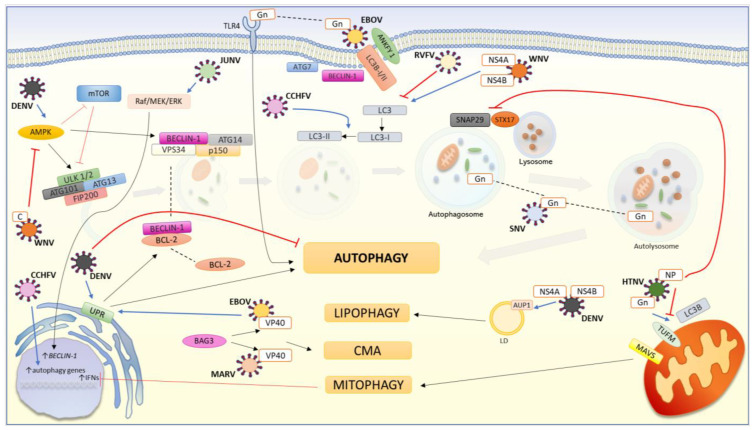
The interaction between viral hemorrhagic fever and autophagic pathway.

**Table 1 cells-11-00871-t001:** Main features of autophagy in viral hepatitis, hemorrhagic fever viruses, and *Lagovirus europaeus* infection.

Virus	Association between the Virus and the Autophagic Pathway	The Role of Autophagy	Reference
Viral hepatitis			
Hepatitis B Virus (HBV)	HBx protein promotes Beclin-1 expression; Hbx protein binds to VPS34; Hbx protein enhances the interaction between VPS34 and TRAF6-Beclin-1 complex; HBx protein dephosphorylates DAPK kinase; HBx protein interacts with c-Myc; SHB protein activates UPR pathway.	HBV induces an autophagic pathway to promote viral replication.	[71,72,73,74,75,76,77,78]
Hepatitis D Virus (HDV)	HDAg protein induces LC3-II accumulation and promotes autophagosome formation; ATG7 and LC3 are involved in HDV replication but ATG5 is crucial.	HDV induces incomplete autophagy to promote viral replication.	[14]
Hepatitis C Virus (HCV)	HCV activates the UPR-dependent autophagy, but the process is incomplete; HCV interferes with Nfr2/ARE pathway via NS3 protein to promote autophagy induction by oxidative stress; p7 protein activates Beclin-1; NS4B protein interacts with Rab5 and Vps34; NS3 binds to IRGM, which activates autophagy; NS5A induces *BECN1* expression; NS5A mediates starvation-induced autophagy; NS5A induces mitochondrial fragmentation, which activates mitophagy.	HCV induces autophagy to promote its replication. Autophagy can negatively regulate HCV replication in the presence of IFN-β.	[82,83,84,85,86,87,88,89]
Viral Hemorrhagic Fever			
Ebola virus (EBOV)	LC3, Beclin-1, and ATG7 are required for EBOV to enter the cell; EBOV induces autophagy in T cells by ER stress; Vp40 protein induces CMA by BAG3.	FAM134B-dependent ER-phagy limits EBOV replication in mouse cells. Activating BAG3-mediated CMA has antiviral properties.	[91,95,96,97]
Marburg (MARV)	Vp40 protein induces CMA by BAG3.	Activating BAG3-mediated CMA has antiviral properties.	[97]
Junin virus (JUNV)	JUNV infection results in an increase in LC3-II level, co-localization of p62/SQSTM1, ATG16, Rab5, Rab7a, co-localization LAMP1 with LC3 and p62/SQSTM1 degradation; NP is associated with the autophagic membrane during infection; JUNV replication requires activation of the Raf /MEK /ERK pathway, which upregulates *Beclin-1*.	JUNV induces an autophagy pathway to promote viral replication. The early stages of autophagosome formation may provide space for JUNV replication	[99,100,133,134]
Hantaan virus (HTNV)	Gn interacts with TUFM and LC3B to induce mitophagy; NP binds to LC3B and SNAP29 to inhibit Gn-induced autophagy.	HTNV induces autophagy to promote viral replication and the production of progeny viruses. HTNV interferes with type I IFN immune responses through mitophagy.	[102,103]
Sin Nombre virus (SNV)	Gn increases LC3-I and LC3-II levels; Gn degrades in the selective autophagic process.	SNV induces autophagy and modulates autophagic machinery to promote its replication and formation of new virions.	[17,105]
Crimean-Congo hemorrhagic fever virus (CCHFV)	CCHFV infection increases *ATG5*, *ATG7*, *ATG3*, *ATG12*, *BECN1 MAP1LC3*, and *p62*/*SQSTM1* transcription; CCHFV causes LC3 lipidation.	Autophagy does not promote or limit viral replication.	[106,107]
Rift Valley fever virus (RVFV)	RVFV infection causes autophagy activation via TLR7 receptor; RVFV infection is associated with reduced LC3B level.	Autophagy plays an antiviral role in the host. The virus may target autophagy to weaken host defense.	[109,110]
Dengue virus (DENV)	DENV uses amphisomes and autophagolysosomes as scaffolds for viral replication; DENV blocks the formation and degradation of autophagosomes and induces p62/SQSTM1 degradation; DENV activates UPR pathway; DENV infection is related with LDs containing viral capsid proteins; DENV activates AMPK kinase; NS4A and NS4B interact with AUP1.	DENV uses some elements of autophagic machinery to viral replication and a path to escape the host’s immune system. DENV induces autophagy to promote its replication and the maturation of infectious particles. Autophagy is an antiviral mechanism in monocytes and *Aedes* mosquito cells.	[112,113,114,115,116,117,118,119,120,121,122,123,124,125]
West Nile virus (WNV)	WNV infection increases the LC3-II level; WNV influences lysosomal colocalization with autophagosomes, LC3B-II turnover, and autolysosomal acidification; NS4A and NS4B influence modifying and aggregating LC3; C protein interacts and mediates AMPK ubiquitination.	Autophagy induced by WNV infection is a host defense mechanism. WNV can inhibit autophagy to promote disease development.	[127,128,129,130,131,132]
Rabbit Hemorrhagic Disease			
*L. europaeus*	*L. europaeus* infection increases LC3-II level, expression of the Atg12-Atg5-Atg16L1 complex, UVRAG and p62/SQSTM1; *L. europaeus* upregulates Beclin-1; *L. europaeus* infection activates UPR pathway.	Further studies are needed.	[68,69,70]

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
