# Peer review of "The Interplay between Autophagy and Virus Pathogenesis—The Significance of Autophagy in Viral Hepatitis and Viral Hemorrhagic Fevers"

_cells, 2022, doi:10.3390/cells11050871_

Round 1

Reviewer 1 Report

Bębnowska and Niedźwiedzka-Rystwej gather in this review the available information on how autophagy and virus causing hepatitis or hemorrhagic fevers are linked. Furthermore, they propose the use of Lagovirus europaeus infection as a safe, convenient research model to study these pathologies.

Overall, I think this is a nice compilation of the subject, and it can be really useful for researchers working in this field. However, some major points hamper the quality of the paper and need to be fixed before it could be considered for publication.

My major concern is the number of typos and writing mistakes that can be found throughout the manuscript. They complicate the reading of the review and mar the quality of the bibliographic compilation. I encourage an extensive revision to fix these problems and ensure that everything is correct and clear.

Some of the mistakes that I found are:

  • Line 37: Change the order of the words. For example, “Xenophagy is a selective form of autophagy targeting viruses”.
  • Line 50-51: Change this sentence so it is easily understood.
  • Line 80: I suggest changing “What is important” to “Specifically”.
  • Line 106: Correct “analyzes” for “analyses”
  • Lines 129 to 131: I suggest rewording this sentence, as the word “problem” is repeated three times in it.
  • Line 132: I suggest changing “, including, high virulence, rare…” to “, including high virulence and rare…”
  • Line 136: I suggest changing “induce” to “induced”.
  • Line 141: I suggest adding parenthesis here, as “in three clinical forms (hyperacute, acute, and subacute) and is…”
  • Line 172: The way it is written is quite complicated; I suggest changing to “membrane, while dissolved cytosolic proteins are transported into vesicles of late endosomal multivesicular bodies in endosomal microautophagy”.
  • Line 176: I suggest changing “what” to “that”.
  • Line 206: I suggest changing “, moreover, it enables” to “, and it enables”.
  • Line 244: There are no “autophagic follicles”, it should be changed to “autophagic vesicles”.
  • Page 6: I suggest changing “from XX h p.i.” to “starting at XX h p.i.”.
  • Lines 246-247: I suggest a simplification of these sentences; for example, “Additionally, the activity of proteins crucial for the autophagy process, like LC2, was assessed in the liver”.
  • Line 255: I suggest moving “upregulation of the beclin-1 gene” to the beginning of this sentence, replacing “it”.
  • Line 330: I suggest adding a colon after “two isoforms”.
  • Line 353: I would clarify the beginning as “HCV can also trigger oxidative stress-induced autophagy”.
  • Line 375: Either the authors add parenthesis to “mitophagy”, or they use this opportunity to explain what “mitophagy” is.
  • Line 431: I suggest removing the punctuation mark after “EBOV”.
  • Line 548: The name of the genes should be italicized: “Atg5, Atg7, and beclin-1”. Please, check this is consistent throughout the manuscript.
  • Line 585: I suggest removing the punctuation mark after “[113]”.
  • Line 600: I suggest rewording this sentence, as I struggle to understand its meaning.
  • Line 613: One or more words appear to be missing in this sentence.
  • Line 616: I suggest removing “appear to be”.
  • Line 649: One or more words appear to be missing in this sentence.
  • Lines 714-715: I suggest removing “that there are many” and rewording the sentence so it is more clear.

This is just a list of the typos and mistakes that I could find, but I suggest the whole text is revised to fix any other additional errors.

Some changes should also be made to the section where the autophagy molecular pathway is described:

  • The authors should explain that ATG stands for “autophagy-related”.
  • Given the fact that the authors try to explain the process from the beginning to the end, talking about the ULK1 or the ATG12-ATG5-ATG16L1 complex (and I appreciate that), I think they should also explain briefly some of the steps that are missing, citing proteins that are involved in the maturation or transportation of the autophagosomes or its fusion to the lysosomes. Some of the proteins that should be cited are ATG10, ATG2, or different protein families like WIPI, SNAREs, or VAMPs.
  • Regarding the second ubiquitin-like protein system, it is important to clarify that this refers to the ATG8 protein family, and LC3 is only one of the different ATG8 orthologues in mammals. Specifically, in humans, there are two subfamilies of ATG8-like proteins: the LC3 and the GABARAP subfamilies. Again, LC3 is just one of them. I think it is important to clarify this, as well as add that there are four different ATG4 orthologues.
  • Similarly to this, SQSTM1/p62 is the most important (or used) autophagy receptor, but there are plenty more, specific for different substrates (either protein aggregates or distinct organelles), Again, while I agree that p62 is the most notorious one in autophagy research, I honestly believe this should be added.

The manuscript feels a little bit repetitive by the end of it, as the authors state too many times that studies analyzing the interplay between autophagy and different virus are limited and that more research is needed. Perhaps just discussing this one or two times, at the beginning and the end of the review, is enough, and it would improve the reading experience.

Regarding the figures, I would like to congratulate the authors as I believe they are really well-done and very informative. I really think they help, and that they are a highlight in this review. However, the quality of the figures is very poor (at least, in the version that I received). Please ensure that the quality of them is good enough whenever this review is published.

I only found a few errors in them:

  • In both figure legends, I suggest changing “locking mechanism” to “blocking mechanism”.
  • In Figure 2, the HCV virus is missing (only its proteins are shown) when showing its effects on mitochondria. Also, this “mitophagy” part is not discussed in the legend.

Would it be possible to add a new figure at the beginning, comparing the similitudes of RHD, VHF, and viral hepatitis? If so, I think it would be extremely helpful, also supporting the proposal of Lagovirus europaeus infection as a suitable research model.

Finally, references 42 and 43 are the same in the “References” list.

Author Response

Dear Reviever,

Thank you for giving us the opportunity to submit a revised draft of our manuscript titled: “The interplay between autophagy and virus pathogenesis - the significance of autophagy in viral hepatitis and viral hemorrhagic fevers “ to Cells. We appreciate the time and effort that you and the reviewers have dedicated to providing your valuable feedback on my manuscript. We are grateful to the reviewers for their insightful comments on our paper. We have been able to incorporate changes to reflect most of the suggestions provided by the reviewers. We have highlighted the changes within the manuscript.

Here is a point-by-point response to your comments and concerns.

Reviewer 1

Bębnowska and Niedźwiedzka-Rystwej gather in this review the available information on how autophagy and virus causing hepatitis or hemorrhagic fevers are linked. Furthermore, they propose the use of Lagovirus europaeus infection as a safe, convenient research model to study these pathologies.

Overall, I think this is a nice compilation of the subject, and it can be really useful for researchers working in this field. However, some major points hamper the quality of the paper and need to be fixed before it could be considered for publication.

My major concern is the number of typos and writing mistakes that can be found throughout the manuscript. They complicate the reading of the review and mar the quality of the bibliographic compilation. I encourage an extensive revision to fix these problems and ensure that everything is correct and clear.

Some of the mistakes that I found are:

Line 37: Change the order of the words. For example, “Xenophagy is a selective form of autophagy targeting viruses”.

Line 50-51: Change this sentence so it is easily understood.

Line 80: I suggest changing “What is important” to “Specifically”.

Line 106: Correct “analyzes” for “analyses”

Lines 129 to 131: I suggest rewording this sentence, as the word “problem” is repeated three times in it.

Line 132: I suggest changing “, including, high virulence, rare…” to “, including high virulence and rare…”

Line 136: I suggest changing “induce” to “induced”.

Line 141: I suggest adding parenthesis here, as “in three clinical forms (hyperacute, acute, and subacute) and is…”

Line 172: The way it is written is quite complicated; I suggest changing to “membrane, while dissolved cytosolic proteins are transported into vesicles of late endosomal multivesicular bodies in endosomal microautophagy”.

Line 176: I suggest changing “what” to “that”.

Line 206: I suggest changing “, moreover, it enables” to “, and it enables”.

Line 244: There are no “autophagic follicles”, it should be changed to “autophagic vesicles”.

Page 6: I suggest changing “from XX h p.i.” to “starting at XX h p.i.”.

Lines 246-247: I suggest a simplification of these sentences; for example, “Additionally, the activity of proteins crucial for the autophagy process, like LC2, was assessed in the liver”.

Line 255: I suggest moving “upregulation of the beclin-1 gene” to the beginning of this sentence, replacing “it”.

Line 330: I suggest adding a colon after “two isoforms”.

Line 353: I would clarify the beginning as “HCV can also trigger oxidative stress-induced autophagy”.

Line 375: Either the authors add parenthesis to “mitophagy”, or they use this opportunity to explain what “mitophagy” is.

Line 431: I suggest removing the punctuation mark after “EBOV”.

Line 548: The name of the genes should be italicized: “Atg5, Atg7, and beclin-1”. Please, check this is consistent throughout the manuscript.

Line 585: I suggest removing the punctuation mark after “[113]”.

Line 600: I suggest rewording this sentence, as I struggle to understand its meaning.

Line 613: One or more words appear to be missing in this sentence.

Line 616: I suggest removing “appear to be”.

Line 649: One or more words appear to be missing in this sentence.

Lines 714-715: I suggest removing “that there are many” and rewording the sentence so it is more clear.

This is just a list of the typos and mistakes that I could find, but I suggest the whole text is revised to fix any other additional errors.

Thank you for all the tips that certainly improved the quality of our manuscript. We made appropriate corrections in all indicated places and checked our manuscript again for grammatical errors and typos.

Some changes should also be made to the section where the autophagy molecular pathway is described:

The authors should explain that ATG stands for “autophagy-related”.

Line 28: The abbreviation "ATG" has been clarified.

Given the fact that the authors try to explain the process from the beginning to the end, talking about the ULK1 or the ATG12-ATG5-ATG16L1 complex (and I appreciate that), I think they should also explain briefly some of the steps that are missing, citing proteins that are involved in the maturation or transportation of the autophagosomes or its fusion to the lysosomes. Some of the proteins that should be cited are ATG10, ATG2, or different protein families like WIPI, SNAREs, or VAMPs.

Lines 210-221 and 255-262: The manuscript was supplemented with the indicated issues.

Regarding the second ubiquitin-like protein system, it is important to clarify that this refers to the ATG8 protein family, and LC3 is only one of the different ATG8 orthologues in mammals. Specifically, in humans, there are two subfamilies of ATG8-like proteins: the LC3 and the GABARAP subfamilies. Again, LC3 is just one of them. I think it is important to clarify this, as well as add that there are four different ATG4 orthologues.

Lines 241-245: The manuscript was supplemented with the indicated issues.

Similarly to this, SQSTM1/p62 is the most important (or used) autophagy receptor, but there are plenty more, specific for different substrates (either protein aggregates or distinct organelles), Again, while I agree that p62 is the most notorious one in autophagy research, I honestly believe this should be added.

 Lines 249-254: The manuscript was supplemented with the indicated issues.

The manuscript feels a little bit repetitive by the end of it, as the authors state too many times that studies analyzing the interplay between autophagy and different virus are limited and that more research is needed. Perhaps just discussing this one or two times, at the beginning and the end of the review, is enough, and it would improve the reading experience.

As suggested, sections on the need for further research have been removed from most subsections. We left this information only in those sections where we considered it appropriate. In addition, we have supplemented the summary section (lines 732-738).

Regarding the figures, I would like to congratulate the authors as I believe they are really well-done and very informative. I really think they help, and that they are a highlight in this review. However, the quality of the figures is very poor (at least, in the version that I received). Please ensure that the quality of them is good enough whenever this review is published.

Thank you for your feedback on our figures. As suggested, the quality of the figures has been improved.

I only found a few errors in them:

In both figure legends, I suggest changing “locking mechanism” to “blocking mechanism”.

In Figure 2, the HCV virus is missing (only its proteins are shown) when showing its effects on mitochondria. Also, this “mitophagy” part is not discussed in the legend.

All errors regarding figures and their descriptions have been corrected as suggested.

Would it be possible to add a new figure at the beginning, comparing the similitudes of RHD, VHF, and viral hepatitis? If so, I think it would be extremely helpful, also supporting the proposal of Lagovirus europaeus infection as a suitable research model.

As recommended, we have added a new figure.

Finally, references 42 and 43 are the same in the “References” list.

The list of references has been updated.

Again, we would like to thank you for your time, expertise, and effort in correcting our paper, which improved due to the changes you have proposed. We hope that now it fulfills the requirements to be published in Cells.

Kind regards,

Paulina Niedźwiedzka-Rystwej

Dominika Bębnowska

Reviewer 2 Report

The review has an original character, but the writing needs no more formality. It is important to note that the entire work is a review. At times the authors put the information as their own opinion, which is not consistent with a review. In certain moments, the description of the results of the articles chosen for the review are very descriptive, which makes the reading heavy. Also, English needs to be improved.

The title should be more direct

Abstract

Line 11- . “Nevertheless, the marriage between autophagy and viruses is a double-edged sword – this process may be used as a strategy to fight with a 12 virus, but also in favor of the virus replication”- Use more formal terms. Remove "marriage" and "double-edge"

Keywords- use words that are not in the title

Introduction

Line 24- in response to the microenvironment, but also physiologically. Quote this.

Line 66- Regardless of the focus of the work initially being on the model, it is entirely a review. Rewrite this sentence.

Line 162- With the description of the effects on infections in the model, it is evident that it is a good research model for hemorrhagic viruses. Please, rewrite the sentence removing the part "our opinion", because it is not based on the authors' opinion that the proposed model is good, but based on what has already been observed in the literature.

Line 167- On the physiology of homeostasis maintenance, add this information to the introduction.

Figure 1 was taken from elsewhere? There is a need to improve the definition of the image and increase the fonts of the subtitles, as they are barely visible.

Line 246- repetitive text

Figure 2 is also without definition.

Line 566- italicize Aedes

Figure 3 is also needs to be improved.

Author Response

Dear Reviewer,

Thank you for giving us the opportunity to submit a revised draft of our manuscript titled: “The interplay between autophagy and virus pathogenesis - the significance of autophagy in viral hepatitis and viral hemorrhagic fevers “ to Cells. We appreciate the time and effort that you and the reviewers have dedicated to providing your valuable feedback on my manuscript. We are grateful to the reviewers for their insightful comments on our paper. We have been able to incorporate changes to reflect most of the suggestions provided by the reviewers. We have highlighted the changes within the manuscript.

Here is a point-by-point response to your comments and concerns.

Abstract

Line 11- . “Nevertheless, the marriage between autophagy and viruses is a double-edged sword – this process may be used as a strategy to fight with a 12 virus, but also in favor of the virus replication”- Use more formal terms. Remove "marriage" and "double-edge"

The text has been changed as suggested.

Keywords- use words that are not in the title

In our opinion it is impossible to omit such words as autophagy or viral hemorrhagic fever, that is why we decided to leave the keywords as they were in the first version of the manuscript, as in this manner they seem to be most informative. We do hope that the Reviewer will understand.

Introduction

Line 24- in response to the microenvironment, but also physiologically. Quote this.

Text has been added.

Line 66- Regardless of the focus of the work initially being on the model, it is entirely a review. Rewrite this sentence.

The text has been changed as suggested.

Line 162- With the description of the effects on infections in the model, it is evident that it is a good research model for hemorrhagic viruses. Please, rewrite the sentence removing the part "our opinion", because it is not based on the authors' opinion that the proposed model is good, but based on what has already been observed in the literature.

The text has been changed as suggested.

Line 167- On the physiology of homeostasis maintenance, add this information to the introduction.

Corrections applied.

Figure 1 was taken from elsewhere? There is a need to improve the definition of the image and increase the fonts of the subtitles, as they are barely visible.

All the figures are original. The figure quality has been improved.

Line 246- repetitive text

Corrections applied.

Figure 2 is also without definition.

The figure quality has been improved.

Line 566- italicize Aedes

Corrections applied.

Figure 3 is also needs to be improved.

The picture quality has been improved.

Again, we would like to thank you for your time, expertise, and effort in correcting our paper, which improved due to the changes you have proposed. We hope that now it fulfills the requirements to be published in Cells.

Kind regards,

Paulina Niedźwiedzka-Rystwej

Dominika Bębnowska

Round 2

Reviewer 1 Report

I appreciate the response from the authors and the fact that they have addressed my earlier concerns.

The only minor thing I would suggest fixing before publishing this work would be writing the features in new Figure 1 with a capital letter. Besides that, I believe the manuscript is now acceptable for publication. 

Reviewer 2 Report

I recommend the work for publication.